

# COUPLING WRF WITH HEC-HMS AND WRF-HYDRO FOR FLOOD FORECASTING IN TYPICAL MOUNTAINOUS CATCHMENTS OF NORTHERN CHINA

Sheik Umar Jam-Jalloh[1], Jia Liu[1*], Yicheng Wang[1], and Yuchen Liu[2]

[1] State Key Laboratory of Simulation and Regulation of Water Cycle in River Basin, China Institute of Water Resources and Hydropower Research, Beijing 100038, China.

[2] State Key Laboratory of Environmental Criteria and Risk Assessment, Chinese Research Academy of Environmental Sciences, Beijing 100012, China.

*Correspondence to*: jia.liu@iwhr.com; Tel.: +86-(0)10-68781656

**Abstract.** The atmospheric-hydrological coupling systems are essential in flood forecasting because they allow for more improved and comprehensive prediction of flood events with an extended forecast lead time. Achieving this goal relies on a reliable hydrological model system that enhances both rainfall predictions and hydrological forecasts. This study evaluated the potential of coupling the mesoscale numerical weather prediction model, i.e., the weather research and forecasting (WRF) model, with different hydrological modeling systems to improve the accuracy of flood simulation. The fully-distributed WRF-Hydro and a simi-distributed Hydrological Engineering Center-Hydrological Modeling System (HEC-HMS) modeling systems were coupled with the WRF model, and the lumped HEC-HMS model was also adopted using the observed gauge precipitation as a benchmark to test the model uncertainty. Four distinct storm events from two mountainous catchments in northern China characterized by varying spatial and temporal rainfall patterns were selected as case studies. Comparative analyses of the simulated flooding processes were carried out to evaluate and compare the performances of the coupled systems with different complexities. The coupled WRF/HEC-HMS system performed better for long-duration storm events and obtained optimal performance for storm events uniformly distributed both temporally and spatially, as it adapted to more rapid recession processes of floods. However, the coupled WRF/HEC-HMS system did not adequately capture the magnitude of the storm events as it had a larger flow peak error. On the other hand, the fully distributed WRF/WRF-Hydro system performed better for shorter-duration floods with higher flow peaks as it can adapt to the simulation of flash floods. However, the performance of the system became poor as uniformity decreased. The performance of the lumped HEC-HMS indicates some source of uncertainty in the hydrological model when compared with the coupled WRF/HEC-HMS, but a larger magnitude error was found in the WRF output rainfall. The results of this study can help establish an adaptive atmospheric–hydrologic coupling system to improve flood forecasting for different watersheds and climatic characteristics.

**Keywords:** Atmospheric-hydrologic coupling system; WRF rainfall output; HEC-HMS model; WRF-Hydro modeling system; Flood simulation.



## 1 Introduction

Floods are frequent and widespread natural hazards that result in substantial annual losses to human lives and properties worldwide (Jonkman, 2005). Due to climate change, the future is expected to bring more intense precipitation, which might potentially lead to an increase in extreme rainfall-induced flood events and elevated flood risk (Mirza, 2003). Flood forecasting is essential to mitigate the impact of floods by providing timely warnings, and enabling proactive measures, that help to safeguard lives, property, and infrastructure in vulnerable areas (Merz et al., 2020). Improving the ability to predict

flood risks ahead of time is essential in the premise of promoting forecast accuracy. To improve the simulation accuracy and extend the forecast lead time, there is a growing trend in favor of substituting the conventional 'throughfall' with the mesoscale numerical weather prediction (NWP) (Ozkaya, 2023; Trinh et al., 2023; Kaufmann et al., 2003). An effective strategy to do this involves coupling the hydrological model with a high-resolution regional NWP model. This approach has demonstrated capabilities to not only improve the accuracy of flood forecasting but also extend real-time forecast lead time,

compared to conventional flood forecasting that relies only on gauge observations as inputs(Seid et al., 2021).

In recent years, coupling hydrological models with high-resolution NWP models covering different scales has emerged as a promising approach to improve flood simulation (Jasper et al., 2002; Bartholmes and Todini, 2005; Nam et al., 2014; Wu et al., 2014; Cattoën et al., 2016; Li et al., 2017; Wu et al., 2020; Chen et al., 2020; Ming et al., 2020; Dasgupta et al., 2023;

Patel and Yadav, 2023; Bacelar et al., 2023). (Jasper et al., 2002) coupled the WaSim-ETH model with surface observations, forecast data from five high-resolution NWP models, and weather radar data for seven extreme flood events. They concluded that future simulation improvement hinged on the NWP model development. (Bartholmes and Todini, 2005) analyzed the effects of coupling meteorological mesoscale quantitative precipitation forecasts at various scales with the TOPKAPI model to extend the flood simulation horizon. The results highlighted the limited reliability of quantitative simulation precipitation

generated by meteorological models. (Cattoën et al., 2016) coupled the NZLAM regional NWP model at high and low resolutions with the TopNet hydrological model for flood simulation, and their findings indicated the advantage of utilizing a high-resolution convective-permitting lagged ensemble simulation over a lower-resolution large-scale model. (Li et al., 2017) coupled the Weather Research and Forecasting quantitative precipitation forecast (WRF QPF) with the Liuxihe model to extend flood forecasting lead in southern China and found that as the lead time increased, both the accuracy of the WRF QPF

and the flood simulation capability decreased. (Chen et al., 2020) coupled the GRAPE_MESO model, a two-dimensional hydrodynamical flood model, and a rainstorm prediction reconstruction method for urban flood simulation. Their results showed that the coupled modeling system achieved accurate predictions with high resolution and an extended lead time. (Ming et al., 2020) produced high-resolution catchment-scale rainfall-runoff and flood forecasting by coupling an NWP model with a GPU-accelerated hydrodynamic model. The system provided a lead time of 34 hours when weather forecasts

were accessible 36 hours in advance. (Patel and Yadav, 2023) researched the coupling of hybrid ensemble Linear Regression,



the HEC-HMS model, and the Bayesian numerical weather model to simulate hourly reservoir inflows. The results showed how the coupling systems can predict reservoir inflows in the Sabarmati River basin in India.

Although recent studies have been conducted to improve flood forecasting by coupling the NWP model with hydrological models, most do not consider the difference of choosing a fully distributed or a semi-distributed model of different complexities when constructing the coupling system. The complexity of the hydrological model plays an important role in determining the generation of the streamflow, which should not be neglected when establishing and evaluating the atmospheric-hydrological coupling system(Ahmed et al., 2023). A fully distributed model divides a watershed into smaller spatial units, allowing for a detailed representation of the entire area. A semi-distributed model groups similar sub-basins, providing a balance between detail and computational efficiency (valiya veettil et al., 2021). By integrating meteorological data from the NWP models into these hydrological models, it is possible to create a holistic understanding of how meteorological inputs impact the generation of the streamflow. Understanding the source of uncertainties involved in this process and how to eliminate them is also of paramount importance in improving the performance of the atmospheric-hydrological systems for flood forecasting.

The main objective of this study is to evaluate the potential of coupling the mesoscale numerical weather prediction model, i.e., the weather research and forecasting (WRF) model, with different hydrological modeling systems to improve the accuracy of flood forecasting. The fully-distributed WRF-Hydro and a simi-distributed Hydrological Engineering Center-Hydrological Modeling System (HEC-HMS) modeling systems were coupled with the WRF model, and the lumped HEC-HMS model is also adopted using the observed gauge precipitation as a benchmark to test the model uncertainty. Four distinct storm events, characterized by varying spatial and temporal rainfall patterns, were selected as case studies. These events occurred in two mountainous catchments along the Daqing River, where precise flood prediction is urgently required to mitigate the risks associated with construction in northern China's downstream area. The WRF model stands out as the predominant mesoscale NWP model for simulating and forecasting rainfall in hydrology and Water resource-related disciplines (Done et al., 2004; Lo et al., 2008; Liu et al., 2012; Haghroosta et al., 2014; Chawla et al., 2018; Yáñez-Morroni et al., 2018; Huang et al., 2023). The advancement of the WRF-Hydro modeling system, built upon the research on the WRF model, has enhanced the efficiency of utilizing WRF for hydrological simulation(Gochis et al., 2013). This innovation addresses the issue of misalignment between the resolution of the atmospheric model and the hydrological model. In recent years, the development of the WRF-Hydro modeling system by the National Center for Atmospheric Research (NCAR) and its collaboration partners has been coupled with the WRF model in various hydrological research projects (Senatore et al., 2015; Ryu et al., 2017; Wang et al., 2020; Sun et al., 2020; Quenum et al., 2022; Wang et al., 2022; Liu et al., 2023; Naabil et al., 2023). The HEC-HMS model is a widely used hydrological model with a flexible structure to be built in either a lumped or a semi-distributed mode. In recent years, there have also been efforts to couple HEC-HMS with the WRF model (Herath et al., 2016; Givati et al., 2016; Niyogi et al., 2022; Tien Thanh et al., 2023; Ting et al., n.d.). The study of (Herath et





al., 2016) demonstrates the coupled WRF and HEC-HMS to be a helpful tool for the flood warning system in Polgolla Barrage in Sri Lanka. (Givati et al., 2016) coupled the HEC-HMS model using a 3km hourly precipitation generated by the WRF model for flood forecasting in the Mediterranean region. (Niyogi et al., 2022) also coupled the WRF model of 1.5km resolution with the HEC-HMS model and HEC-RAS 2D Hydraulic model of 10m resolution.

This study establishes two atmospheric–hydrological systems by coupling the WRF model with WRF-Hydro and the HEC-HMS model to forecast four typical storm events in the study basin. This study utilizes a "one-way" coupling approach between the two hydrological model structures and the WRF model. This means that WRF output drove the hydrological models without reciprocally influencing the atmospheric modeling processes. The 1x1 km output rainfall from the WRF model is used to drive the WRF-Hydro and the HEC-HMS model models to produce flood forecasting. Also, the lumped

HEC-HMS driven by the observed garage precipitation is used to produce flood forecasting. Comparative analyses were carried out to evaluate the performances of the forecast processes of the four storm events by the coupled atmospheric-hydrological and lumped systems and to identify the source of uncertainties further.

The analyses of the forecast processes were divided into three parts:

- Evaluate and compare the performance of the coupled systems, i.e., WRF/WRF-Hydro and WRF/HEC-HMS, with
different complexities.

- Evaluate and compare the performance of the coupled WRF/HEC-HMS and lumped HEC-HMS model driven by observed rainfall to analyze the model uncertainty.

- Evaluate and analyze the error of the WRF model output rainfall and its resulting uncertainty in the atmospheric-hydrological coupling systems.

**2. Study area and data**

This study adopts the Fuping and Zijingguan sub-catchments within the Daqing River basin, located in northern China, as study areas. Fuping (2219km2) and Zijingguan (1760km2) are typical mountainous sub-catchments located in the upper Shahe River of the southern branch and the upper Juma River of the northern branch of the Daqing River, respectively, as shown in Figure 1. The Fuping sub-catchment has a longitudinal river slope of 5.7% and a residential area of 0.63%. The

Zijigguan sub-catchment has a longitudinal river slope of 5.5% and a residential area of 0.52%. The predominant land use in the Daqing River basin is farmland, forestland, and grassland, with a granitic gneiss type of geology. The Daqing River basin experiences severe soil erosion attributed to dry soil conditions and excessive groundwater exploitation. Additionally, during the storm season, the river undergoes substantial seepage. The mean annual rainfall is approximately 490 mm and 650 mm for Fuping and Zijingguan, respectively, with most rain occurring between late May and early September. Summer storms

with high intensities and short durations are typical of the rainfall found in China's mountainous regions, such as Fuping and Zijingguan. Therefore, they mostly result in severe flood disasters in the downstream Daqing River basin.

.

**Figure 1. Locations of the two study sub-catchments in the Daqinhe catchment.**






## 2.2 Storm Events

Four storm events with 24-hour durations and relatively high flow peaks, as shown in Figure 2, are selected to test the performances of the coupled hydrological rainfall-runoff modeling systems constructed in this study. Three events happened in the Fuping sub-catchment and one in the Zijingguan sub-catchment. Table 1 shows the duration, cumulative rainfall, and

peak discharges of the storm events. The four 24-hour storm events are categorized according to their spatial and temporal distributions. The coefficient of variance ($C_v$) of the storm event was calculated to designate the different homogenous characteristics and is calculated as:

$$C_v = \sqrt{\frac{\sum_{i=1}^{N}(\frac{x_i}{\bar{x}} - 1)^2}{N}}$$                (1)

When calculating $C_v$ for the spatial distribution, the 24-hour cumulative rainfall at any $ith$ rain gauge is $x_i$, the cumulative average rainfall of all stations is $\bar{x}$, and $N$ is the total number of all the stations. For $C_v$, in temporal distribution calculation,

the catchment areal rainfall at any hour $i$ time step is $x_i$, the average areal rainfall of all time steps is $\bar{x}$, and $N$ is the total duration of the storm events (24h).

**Table 1. The Selected four 24-hour Storm Events in Fuping and Zijingguan sub-catchments.**

| Event | Sub-catchment | Start time | End time | 24h cumulative rainfall | Peak discharge ($m^3s^{-1}$) |
|---|---|---|---|---|---|
| 1 | Fuping | 29/07/2007 20:00 | 30/07/2007 20:00 | 63.38 | 29.70 |
| 2 | Fuping | 30/07/2012 10:00 | 31/07/2012 10:00 | 50.48 | 70.70 |
| 3 | Fuping | 11/8/2013 7:00 | 12/8/2013 7:00 | 30.82 | 46.60 |
| 4 | Zijingguan | 21/07/2012 04:00 | 22/07/2012 04:00 | 172.2 | 2580.00 |

The $C_v$ Values of the selected storm events shown in Table 2 reflect the spatio-temporal derivation of the catchment accumulative gauge rainfall of each station and the average rainfall at each time step, respectively. A smaller $C_v$ value indicates that the rainfall distribution is more uniform or even in space or time. As seen in Table 1, storm Event 2, with the smallest spatial distribution $C_v$ value, is the most uniform in space, followed by Event 1, Event 4, and Event 3. The most uniform in time is Storm Event 1, with the smallest temporal distribution $C_v$ value, followed by Event 2, Event 4, and Event

3. Generally, from the categorization, Event 1 has a rainfall that is uniform in both spatial-temporal distributions, and Event





2 has a rainfall that has a uniform spatial distribution but non-uniform temporal distribution. Events 3 and 4 have non-uniform rainfall in both spatial-temporal distributions.

**Table 2. The coefficient of variance ($C_v$) of the four 24-hour storm events.**

| Rainfall event | Spatial distribution | Temporal distribution |
|----------------|----------------------|-----------------------|
| 1 | 0.3975 | 0.6011 |
| 2 | 0.1927 | 1.0823 |
| 3 | 0.7400 | 2.3925 |
| 4 | 0.6098 | 1.8865 |

160

The selected rainfall-runoff storm events have different lengths and flood recession times. We employ a 71-hour duration for events 1 and 3, a 67-hour duration for event 2, and a 36-hour duration for event 4, as shown in Figure 2. Event 4 can be noted as an extreme situation; it has larger $C_v$ values and occurred in a shorter duration with the highest flow peak and rainfall intensity. Event 4 corresponds to the events at the most significant monitoring point that occur once every 500 years

165 and is regarded as one of the largest flood disasters. These different rainfall-runoff characteristics are used in the evaluation of the simulation results.

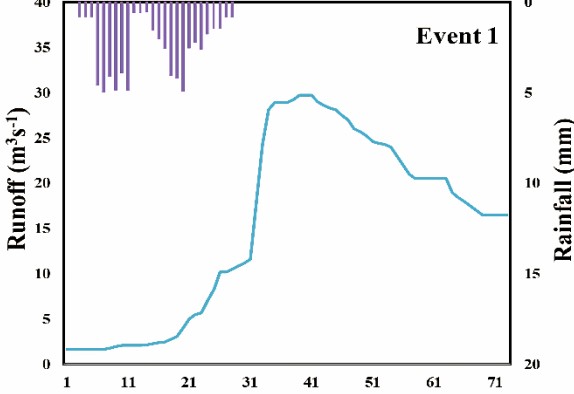

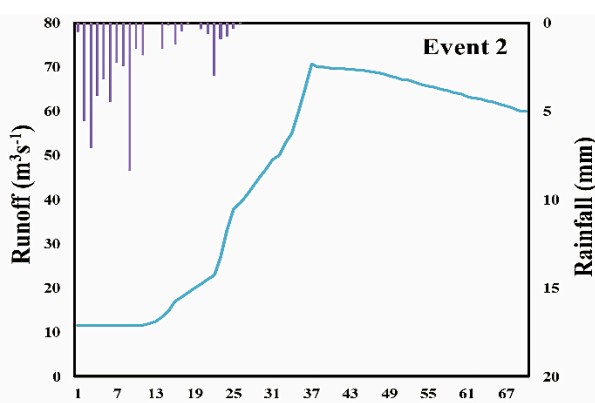





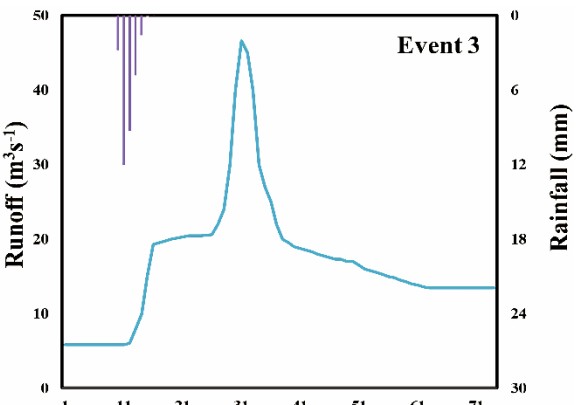
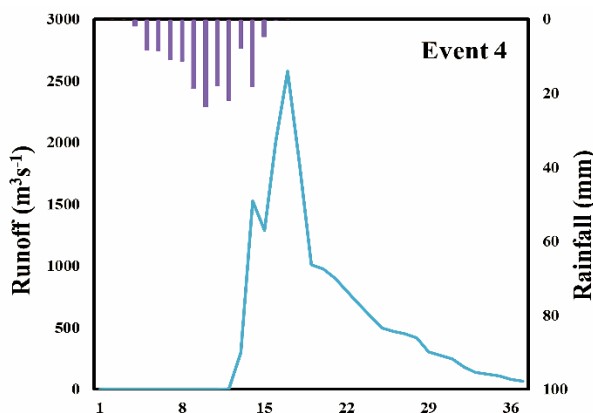

**Figure 2. The rainfall-runoff observations of the four 24-hour storm events.**

## 3. Hydrological models and Experimental design

### 3.1 WRF Model Setup

The Weather Research and Forecasting (WRF) non-hydrostatic model is a widely used numerical weather prediction system that simulates and forecasts atmospheric processes (Powers et al., 2017). Its structure includes initialization, dynamics, physics, grid options, output tools, boundary conditions, nesting capabilities, and parallel computing support, making it versatile for various applications and domains. The versatility and flexibility abilities of the WRF model predict weather patterns at different spatial and temporal scales, from global to regional and even local levels (Cassola et al., 2015). Additional details about the WRF model can be explored in (Skamarock and Klemp, 2008).

This study employs the most widely used parameter setup option from previous studies in northern China because determining the optimal parameters for future storm events poses a significant challenge(Tian et al., 2017b). More information on the parameter setup can be explored in (Tian et al., 2017a). Table 3 shows the main parameterization configurations of the WRF model for the two sub-catchments that have more influences on precipitation generation. The initial boundary conditions for simulation are derived from the 1˚x1˚ FNL driving data at 6-hour intervals, with the integration time step set at 6 seconds (Zhu et al., 2022). The FNL data is the Final Operational Global Analysis meteorological data (available at: http://rda.ucar.edu/datasets/ds083.2/). The WRf output data interval is set at 1 hour with a spin-up time of 6 hours. Three nested domains are set up over the Fuping and Zijingguan sub-catchments. The innermost domain of the WRF is set up at a 1 km horizontal resolution and the nesting ratio of the three layers configured at 1:3. The grid center of the Fuping sub-catchment is at 39°04′15″N and 113°59′26″E, and the nesting grid division from domain 1 to domain 3 are 252 × 234 km², 144 × 126 km² and 96 × 84 km². The grid center of the Zijingguan sub-watershed is at 39° 25′



59″N and 114° 46′01″E, and the nesting grid division from domain 1 to domain 3 are 216 × 198 km², 108 × 90 km², and 72 × 42 km². With a Lambert projection, a 40 vertical discretization up to a 50hPa top-layer pressure is set up for the three nested domains (Tian et al., 2020). The downscaled output precipitation from the WRF model serves as input to drive both the HEC-HMS and the WRF-Hydro models.

**Table 3. Main WRF model parameterization configurations for the two sub-catchments.**

| Parameterization | Chosen Option |
| --- | --- |
| Driving data | FNL at 6h |
| Integration time step | 6s |
| Output  interval | 1h |
| Fuping sub-catchment Grid center | 39°04′15″N, 113°59′26″E |
| Zijingguan sub-catchment Grid center | 39° 25′ 59″N, 114° 46′01″E |
| Nesting ratio | 1:3 |
| Horizontal resolution | Dom1: 9km |
| | Dom2: 3km |
| | Dom3: 1km |
| Fuping  nesting grid division | Dom1: 252 × 234 km² |
| | Dom2: 144 × 126 km² |
| | Dom3: 96 × 84 km² |
| Zijingguan nesting grid division | Dom1: 216 × 198 km² |
| | Dom2: 108 × 90 km² |
| | Dom3: 72 × 42 km² |
| Projection resolution | Lambert |
| Vertical discretization | 40 layers |
| Pressure | 50hPa |

## 3.2 WRF-Hydro model

The Weather Research and Forecasting Hydrological (WRF-Hydro) model is a widely used, fully distributed hydrological modeling system that integrates atmospheric, land surface, and hydrological processes to simulate and predict surface and subsurface water fluxes. Its structure is designed to represent the complex interactions between the atmosphere and the land surface, including precipitation, runoff, streamflow, and soil moisture. WRF-Hydro model can only be run by coupling with the WRF model or utilizing meteorological data to establish an atmospheric-hydrological model system. This study





implements a one-way run utilizing the WRF-Hydro modeling system version 3.0 with the WRF model (Gochis et al., 2015). The schematic structure of the one-way coupling of the WRF/WRF-Hydro model is shown in Figure 3.


Key components of the WRF-Hydro model structure include:

➢ Atmospheric Component (WRF): The WRF model provides the hydrological component for the coupled WRF/WRF-Hydro system with meteorological inputs such as temperature, wind, precipitation, and other meteorological variables.

➢ Land Surface Model (LSM): The LSM component simulates processes occurring at the land surface, such as evapotranspiration, infiltration, and surface runoff. It considers various land cover types and soil properties to model how water moves and interacts with the land.

➢ Spatial Domain: WRF-Hydro operates on a spatial grid, representing hydrological processes at various scales, from small catchments to large river basins.

➢ Hydrological Component: This is the core of WRF-Hydro, where the hydrological processes are simulated. It includes several sub-components, such as:

- Routing: Simulates water movement through river networks and channels, accounting for flow routing and storage dynamics. WRF-Hydro uses a simplified Muskingum-Cunge routing equation for river routing:

$$\frac{\partial Q}{\partial t} = \frac{1}{\Delta x}(Q^n - Q^{n-1}) - \frac{1}{2}\left(\frac{S^n + S^{n-1}}{2}\right)\left(\frac{\Delta Q^n}{\Delta t}\right) \tag{2}$$

Where, $Q$ is the river discharge, $t$ is time, $\Delta x$ is the river reach length, $S$ is the channel storage, $n$ and $n-1$ represent the current and previous time steps, $\Delta Q$ is the change in discharge over time $\left(\frac{\partial Q}{\partial t}\right)$.

- Runoff generation: The WRF-Hydro model generates runoff using a simple water balance (SWB) method. The topsoil layer experiences a surface infiltration excess when the precipitation capacity surpasses the infiltration capacity, resulting in a corresponding alteration in the surface water depth $h$ (m):


$$\frac{\partial h}{\partial t} = \frac{\partial p_e}{\partial t}\left\{1 - \frac{[\sum_{K=1}^{4} Z_i(\delta_s - \delta_k)]\left[1 - \exp\left(-s\frac{R_{dt}}{R_{fd}}\frac{\Delta t}{86400}\right)\right]}{p_e + [\sum_{k=1}^{4} Z_i(\delta_s - \delta_k)]\left[1 - \exp\left(-S\frac{R_{dt}}{R_{fd}}\frac{\Delta t}{86400}\right)\right]}\right\} \tag{3}$$

Where, $h$(m) represents the change in surface water depth, $k$ is an integer representing the soil layer (ranging from 1 to 4), $\delta_k$ (m³m⁻³) and $Zk$ (m) are the soil moisture grid and depth of the $k$th soil layer, $\delta_s$





(m³m⁻³) represents the maximum soil moisture content, $\Delta t$ (s) represents the model time step, $S$ denotes the coefficient from the regulating runoff infiltration Richards' equation, and $R_{fd}$ and $R_{dt}$ denote the saturated hydraulic conductivity and the tunable coefficients for surface infiltration, respectively.

- Groundwater Flow: This represents water movement through subsurface aquifers, which can significantly influence streamflow. The Baseflow module typically describes groundwater flow. For example, the Bucket model uses a conceptual storage equation. This equation depicts groundwater storage changes over time, influenced by input recharge and outflow proportional to the difference between current storage and baseflow threshold:

$$\frac{dS}{dt} = R - K_b \cdot (S - S_0) \tag{4}$$

Where, $\frac{ds}{dt}$ is the rate of change of storage with respect to time, $R$ is the recharge to the groundwater storage (from excess soil moisture), $K_b$ is the baseflow recession coefficient, $S$ is the storage of groundwater, $S_0$ is the baseflow threshold storage (also known as initial or critical storage).

The horizontal resolution of WRF-Hydro is specified by segmenting the inner domain of WRF into a grid spacing of 100 m in this study. The Noah-MP LSM contains four soil layer thicknesses, 10cm, 30cm, 60cm, and 100cm, with a soil column of 2m from top to bottom. Enabling the fully coupled option initiates the involvement of the hydrological module, disaggregation-aggregation module, and Land Surface Model (LSM) components of WRF-Hydro when running WRF. It should be noted that the default Noah configurations in WRF-Hydro were employed rather than using site-specific settings. Also, the baseflow bucket model is switched off for simulation periods; the WRF-Hydro model primarily accumulates sub-surface runoff and redistributes it to the channel, effectively increasing river flow (Xue et al., 2000). It should be noted that the findings presented in this study should be considered a benchmark for the WRF-Hydro fundamental model performance. The intention is to offer valuable insights for future users of the model operating in particular basins within northern China and comparable regions, as well as to provide guidance for prospective model enhancements in future years.


**Figure3. The schematic structure of the one-way coupled WRF/WRF-Hydro model** (Verri et al., 2017)**.**

255 **3.3 HEC-HMS model**

The Hydrologic Engineering Center's Hydrologic Modeling System (HEC-HMS) is a comprehensive software tool developed by the United States Army Corps of Engineers for hydrological modeling and the simulation of watershed runoff. It is used for a wide range of applications, including flood forecasting, reservoir management, and water resource planning. The HEC–HMS  model Version 4.10 is employed in this study (Bartles et al., n.d.). Compared to older versions, this
260  upgraded version allows for the import of grid data.

The HEC-HMS model structure generally consists of various components:



> ➤ Data Processing: HEC-HMS includes tools like GIS connection, data import, etc, for data processing and manipulation. They typically involve watershed delineation, interpolation, time step adjustments, and data transformations.

> ➤ Metrologic data: HEC-HMS allows the input of historical or synthetic rainfall data of various formats (time series or gridded).

> ➤ Hydrologic models: The HEC-HMS provides a range of hydrological models for simulating the precipitation-to-runoff transformation, such as:

> • Loss Model: HEC-HMS includes methods to estimate losses due to interception, depression storage, and infiltration. The direct runoff is calculated as follows:

$$(Q) = \frac{(P - I_a)^2}{(P - I_a + S)}$$
(5)

Where, $Q$ is direct runoff, $P$ is precipitation, $I_a$ is initial abstraction, and $S$ is potential maximum retention after runoff begins.

> • Routing Model: HEC-HMS employs various routing methods. For example, the Muskingum-Cunge method uses the following equation to route flow through river channels:

$$Q(k + 1) = (1 - x) \times Q(k) + x(P - P_{loss})$$
(6)

Where, $Q(k + 1)$ is the outflow at the next time step, $Q(k)$ is the current outflow, $x$ is the routing parameter, $P$ is inflow, and $P_{loss}$ represents losses. More information about the Muskingum-Cunge method can be found in (Niazkar and Zakwan, 2022).

> ➤ Unit Hydrograph: The HEC-HMS model uses the unit hydrograph model to distribute rainfall in time or space. It describes the relationship between rainfall excess and direct runoff. This includes various methods for hydrograph generation, such as:

> • ModClark Unit Hydrograph Method: It is a modified version of the Clark Unit Hydrograph method. Additional details on the Clark unit hydrography can be explored in (Che et al., 2014). The Modclark uses a simple linear convolution equation:

$$Q(t) = \frac{P(t)}{A} \times UH(t)$$
(7)


Where, $Q(t)$ is the discharge (runoff) at time $t$, $P(t)$ is the precipitation at time $t$, $A$ is the area of the watershed, and $UH(t)$ is the Modclark unit hydrograph at time $(t)$.


- SCS (Soil Conservation Service) Unit Hydrograph Method: The SCS method employs a time-area approach to develop a unit hydrograph that represents the response of a watershed to a unit depth of excess rainfall. More information about the SCS Unit Hydrograph can be found in (Shatnawi and Ibrahim, 2022). The following equation represents the SCS Unit Hydrograph:

$$Q(t) = \frac{P(t)}{12} \times UH\ (t) \tag{8}$$

Where, $Q(t)$ is the discharge (runoff) at time $t$, $P(t)$ is the precipitation at time $t$, $UH(t)$ is the SCS unit hydrograph at time $t$.

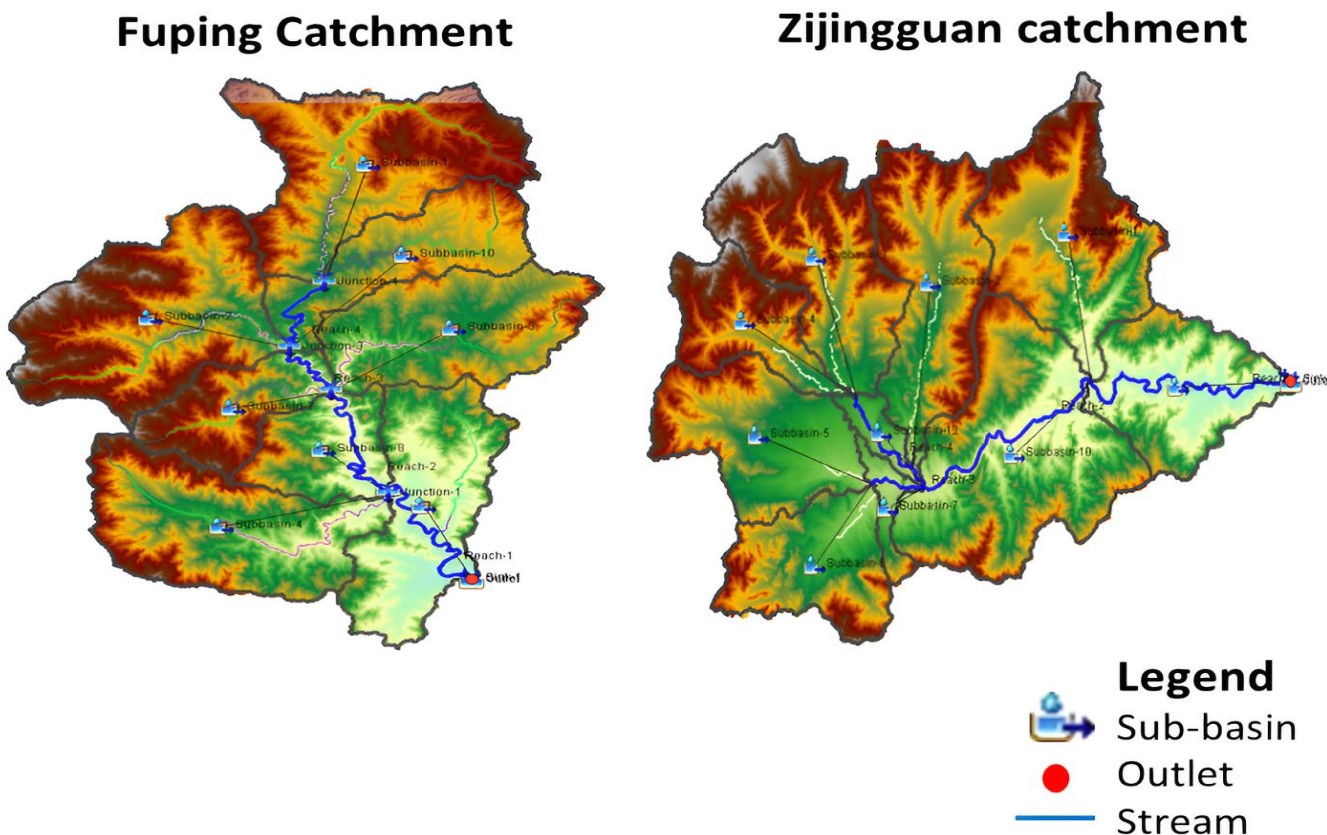

**Figure 4. HEC-HMS schematic DEM map of Fuping and Zijingguan catchments.**





## 3.4 Experimental design

In atmospheric-hydrological coupling, two-way coupling is essential for studying complex climate interactions, while the one-way coupling is often employed for practical meteorological or climate prediction models to simplify computational demands and focus on specific phenomena. Therefore, this study adopts the one-way coupling when constructing the atmospheric-hydrological coupling systems. The experimental structure for the gridded coupled atmospheric-hydrological systems and the lumped system, which comprise three types of hydrological models, i.e., WRF, WRF-Hydro, and HEC-

HMS models, is shown in Figure 5. The WRF model is used to downscale the 1°x 1° FNL data, which will generate gridded rainfall to drive the HEC-HMS and the WRF-Hydro model for flood simulation. The lumped HEC-HMS model is also used for flood simulation using the observed rainfall. The purpose is to set a benchmark for the coupled WRF/HEC-HMS in order to analyze the source of model uncertainties in the coupled atmospheric-hydrologic system.

When calibrating the WRF-Hydro model, careful consideration was taken for several crucial parameters that have the potential to impact flood forecasting accuracy significantly. These parameters were identified through prior research on parameter sensitivity analysis conducted in the study region (Liu et al., 2021). These parameters include the scaling parameter for overland flow roughness (OVROUGHRTFAC), the scaling parameter for surface retention depth (RETDEPRTFAC), the channel Manning roughness parameter (MannN), and the parameter for runoff infiltration

(REFKDT).

For the calibration of the HEC-HMS model, the studied watersheds were delineated into 8 and 11 sub-basins for Fuping and Zijingguan, respectively, as shown in Figure 4. The 1x1 km output gridded rainfall from the WRF was imported and interpolated using the bilinear resampling method. The Modclark and Soil Conservation Service (SCS) unit hydrograph

methods were employed to simulate excess precipitation into direct surface runoff for the gridded and lumped, respectively. The initial and constant method was employed to model infiltration loss, and the exponential recession model was used to model baseflow for both gridded and lumped HEC-HMS. Typically, in standard situations, the model calibration process involves making subjective adjustments to its parameters through a trial-and-error approach. While it is possible to calibrate the model manually, HEC-HMS additionally provides an inherent automatic optimization procedure designed to assess the

suitability and practicality of parameter values and their respective ranges for the intended use of the model.

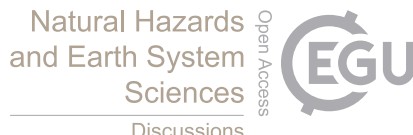

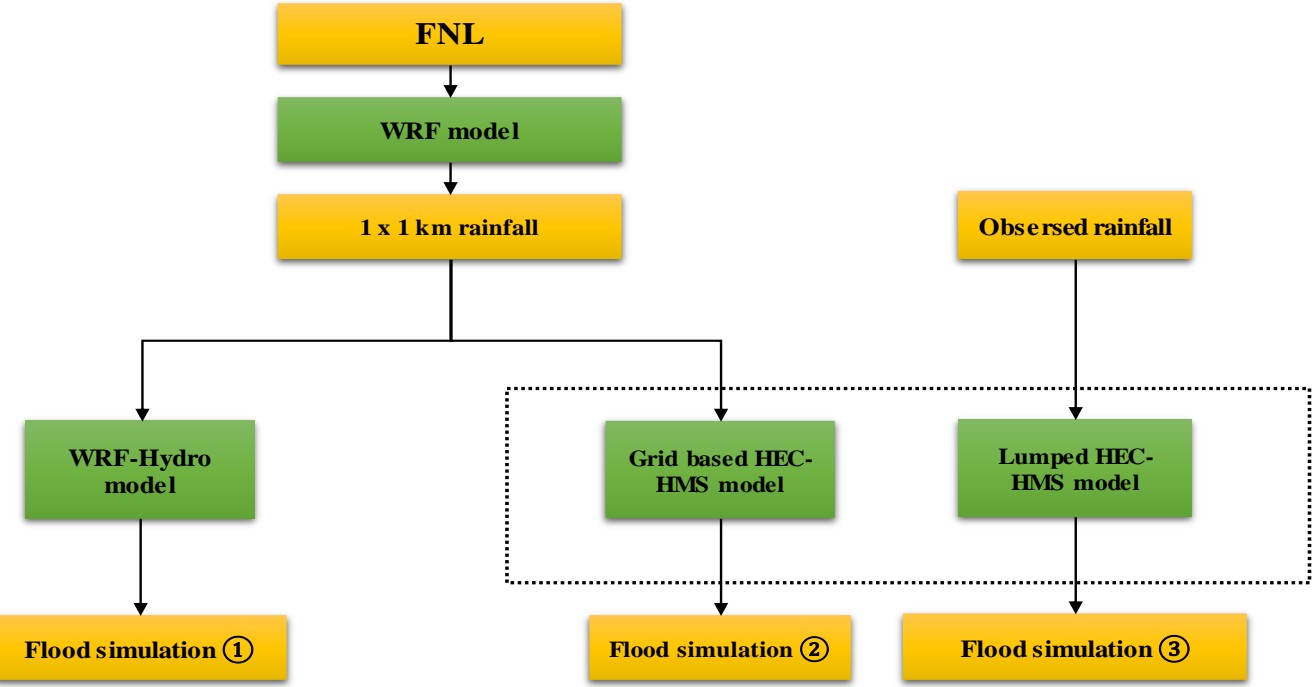

**Figure 5. Experimental structure for the coupled atmospheric-hydrological and lumped modeling system.**

Comparative analyses of the forecast flood processes were carried out to evaluate and compare the performances of the coupled and lumped systems. The flood forecasting results are evaluated using the following statistical criteria: The Nash efficiency coefficient (NSE), root mean square error (RMSE), relative flood peak ($R_f$), and the relative flood volume ($R_v$).

$$NSE = 1 - \frac{\sum_{i=1}^{N}(y_i' - y_i)^2}{\sum_{i=1}^{N}(y_i - \overline{y})^2} \tag{9}$$

$$RMSE = \sqrt{\frac{1}{N}\sum_{i=1}^{N}(y_i' - y_i)^2} \tag{10}$$

$$R_f = \frac{(y_f' - y_f)}{y_f} \tag{11}$$





$$R_v = \frac{(y_v' - y_v)}{y_v} \tag{12}$$

Where, $y_i'$ and $y_i$ represents the simulated and observed discharge flow at a specific time step denoted by $i$, $N$ denotes the total number of flood event time steps, $\bar{y}$ is the calculated average discharge, $y_f'$ and $y_f$ represents the simulated and observed flood peaks, respectively, and $y_v'$ and $y_v$ represents the simulated and observed flood volumes, respectively.

## 4. Results

### 4.1 Results from the coupled WRF/HEC-HMS and WRF/WRF-Hydro systems


The simulation results of the coupled WRF/HEC-HMS and WRF/WRF-Hydro modeling systems are shown in Figure 6 and Table 4. As demonstrated in section 3.4, the 1x1 km output rainfall from the WRF model is used to simulate the four storm events using the coupled atmospheric-hydrological systems. Comparing the simulation results, it can be seen that the coupled WRF/HEC-HMS model performs better for storm events 1 and 3 with NSE of 0.84 and 0.79, respectively, as these events

have longer durations, demonstrating that the model adapted well in modeling prolonged floods. Also, the coupled WRF/HEC-HMS best performance is Event 1 (NSE= 0.84), characterized by long duration and relatively small flood magnitude, demonstrating the ability to model floods subjected to rapid recession. The coupled WFR/WRF-hydro performs better for events 2 and 4 with NSEs of 0.63 and 0.62, respectively, which are floods with very high magnitudes and shorter durations, demonstrating abilities in modeling flash floods.


**Table 4. Simulation results of the coupled WRF/HEC-HMS and WRF/WRF-Hydro system for the four storm events**

| Storm event | $R_f$ (%) | | RMSE | | NSE | |
|---|---|---|---|---|---|---|
| | HEC-HMS | WRF-Hydro | HEC-HMS | WRF-Hydro | HEC-HMS | WRF-Hydro |
| Event 1 | -22.56 | -10.43 | 4.26 | 7.11 | 0.84 | 0.56 |
| Event 2 | -16.12 | -5.75 | 14.13 | 16.06 | 0.51 | 0.63 |
| Event 3 | -10.73 | -12.77 | 3.90 | 6.89 | 0.79 | 0.34 |
| Event 4 | -58.48 | -54.90 | 404.55 | 513.59 | 0.58 | 0.62 |
| | | | | | | |
| Average | -26.97 | -20.96 | 106.71 | 135.91 | 0.68 | 0.54 |

By comparing the observed and simulated hydrographs of the coupled WRF/HEC-HMS and WRF/WRF-Hydro modeling systems (Figure 6), it can be seen that the flow peaks were underestimated for both models. WRF/HEC-HMS has the largest





flow peak error, ranging from -10.78% to -58.48%, but a better RMSE ranging from 4.26 to 404.55. The worst flow peak
error (-58.48%) was for event 4 (highest flow peak and short duration), and It only has a better flow peak error for storm
event 3 (-10.73%) when compared to the coupled WRF/WRF-Hydro. The WRF/WRF-Hydro has a  better flow peak error
ranging from -5.75% to -54.90% but higher RMSE ranging from 7.11 to 513.59. Its worst flow peak error was also for event
4 (-54.9%), for which it had a better performance. However, the coupled WRF/HEC-HMS model with an NSE value ranging

from 0.51 to 0.84 and the coupled WRF/WRF-Hydro with an NSE value ranging from 0.34 to 0.63 indicates that the HEC-
HMS model results in a better average performance when coupled with WRF.

**Figure 6. Simulated flood hydrographs of the coupled WRF/HEC-HMS and WRF/WRF-Hydro systems for the four storm events.**





## 4.2 Results from the Lumped HEC-HMS Model driven by the observed rainfall

Considering the unsatisfactory performance of the coupling systems, the four storm events were also simulated with the lumped HEC-HMS using the observed rainfall (demonstrated in section 3.4 ). Lumped models do not possess the necessary spatial data to depict hydrological processes accurately because they only have temporal inputs. The simulation results of the lumped HEC-HMS are shown in Table 5 and Figure 7.

**Table 5. Simulation results of the lumped HEC-HMS model driven by observed rainfall for the four storm events.**

| storm Events | $R_f$ (%) | RMSE | NSE |
|---|---|---|---|
| Event 1 | -1.01 | 3.19 | 0.90 |
| Event 2 | -7.78 | 14.77 | 0.59 |
| Event 3 | 0.86 | 6.11 | 0.48 |
| Event 4 | -10.77 | 279.14 | 0.81 |
| Average | -4.68 | 75.80 | 0.70 |

As shown in Table 5, the best performance of the lumped HEC-HMS is event 1 with NSE = 0.90, which has the best uniform temporal distribution with a long duration. The worst performance is found in event 3 with NSE= 0.48, which has the least uniform temporal distribution. Generally, the lumped HEC-HMS obtained a late flood peak for some storm events, i.e., -9h

and -1h for events 1 and 3, respectively. Also, it underestimated the flow peak for events 2 and 4, indicating some uncertainty in the model. The lumped HEC-HMS model shows an average performance that is not too great compared to the coupled WRF/HEC-HMS system, i.e., (Average Lumped HEC-HMS NSE value) 0.70 – 0.68 (Average coupled WRF/HEC-HMS) = 0.02.

**Figure 7. Flood forecasting hydrographs of the coupled WRF/HEC-HMS and Lumped HEC-HMS systems for the four storm events.**

The simulation results ($R_f$, RMSE, NSE) of the coupled WRF/HEC-HMS and lumped HEC-HMS model were compared (Rp-Rp-lumped, RMSE-RMSE-lumped, NSE-NSE-lumped ) as shown in Figure 7 and Table 6. This comparison analyses the level of uncertainty in the hydrological model. The largest error in performance was found in event 3 (NSE = 0.31), which has the largest non-uniform temporal distribution, showing a better performance for the coupled WRF/HEC-HMS. For the other three events, the difference between the coupled WRF/HEC-HMS and lumped HEC-HMS model increased as the rainfall temporal distribution heterogeneity increased. This shows that when the rainfall temporal distribution is uniform, as in event 1 < event 2 < event 4, the performance error became lesser -0.07, -0.08, and -0.22, respectively. Also, there is a large difference in flow peak error and RMSE, indicating the level of underestimation of the flow peak from the coupled WRF/HEC-HMS, more notably in storm event 4. These comparisons indicate some uncertainty from the coupled WRF/HEC-HMS model.





**Table 6. Comparison between the coupled WRF/HEC and lumped HEC-HMS model for flood forecasting of the four storm events.**

| storm Events | $R_f$ | $R_f$- $R_{f\text{-lumped}}$ | RMSE | RMSE-RMSE$_{\text{-lumped}}$ | NSE | NSE-NSE$_{\text{-lumped}}$ |
|---|---|---|---|---|---|---|
| Event 1 | -22.56 | 21.55 | 4.26 | 1.07 | 0.84 | -0.07 |
| Event 2 | -16.12 | 8.35 | 14.13 | 0.64 | 0.51 | -0.08 |
| Event 3 | -10.73 | 9.87 | 3.90 | 2.22 | 0.79 | 0.31 |
| Event 4 | -58.48 | 47.71 | 404.55 | 125.41 | 0.58 | -0.22 |

**4.3 Error in the simulated WRF rainfall**

The simulated WRF rainfall is analyzed to determine the error level, which influences the forecasting results of the coupled systems. The relative errors ($R_v$) of the 24-hour rainfall accumulations of the observed and simulated four storm events are shown in Table 7. Figure 8 displays the temporal variations of the observed and simulated rainfall in the accumulative curves and time series bars for the four storm events.


**Table 7. The four 24-hour storm events observed and simulated WRF rainfall.**

| Storm event | Observations (mm) | WRF simulations (mm) in the innermost domain | RE (%) |
|---|---|---|---|
| Event 1 | 63.38 | 69.42 | 9.53 |
| Event 2 | 50.48 | 36.88 | -26.94 |
| Event 3 | 30.82 | 14.90 | -51.66 |
| Event 4 | 172.2 | 57.29 | -66.73 |

It can be seen in Table 7 and Figure 8 that Event 1, which has the most uniform temporal distribution with $C_v$ = 0.60, as shown in Table 2, has a better simulation result compared to the other storm events, having the lowest $R_v$ value of 9.53%.

The largest relative errors were found in storm events 3 and 4, with the most non-uniform temporal rainfall distributions. Generally, both coupled systems performed poorly, with the largest flow peak error for storm event 4 having the largest relative error of -66.73%. Therefore, it can be seen that the simulation uncertainty of the storm events is directly proportional to the temporal rainfall non-uniformity, especially for events with very high peaks.





Figure 8: The 24h accumulative curves of the observed and the simulated rainfall of the four storm events.

The spatial variation of the 24-hour accumulations of rainfall of the four storm events is further analyzed in the study catchments. Figure 9 shows the spatial patterns of the accumulation rainfall distribution from the observed rain gauges, simulated WRF output, the coupled WRF/WRF-Hydro, and the spatial differences of the accumulation rainfall distribution of simulated WRF output and the coupled WFR/WRF-Hydro model (i.e., coupled WRF/WRF-hydro minus WRF), respectively. As seen in the figures, the largest spatial variation was found in Event 3, with the most non-uniform spatial distribution $C_v = 0.74$ (Table 2) compared to the other events. The simulations of the coupled WRF/WRF-Hydro system exhibited more noticeable variations for event 3 compared to the WRF model. This spatial difference is evident in subfigure (c) for Event 3, Illustrating a better performance of the coupled WRF/WRF-Hydro model compared to the WRF model at the crucial storm center. This improvement is attributed to the model's efficient rainfall spatial redistribution.





Storm event Event 4, characterized by intense convective rainfall, experienced a rapid and substantial increase in rainfall intensity within a shorter duration, with maximum gauge cumulative values reaching 355 mm. Both the WRF model and the coupled WRF/WRF-Hydro system underestimated this storm, with the WRF model exhibiting even poorer performance.

This suggests a failure to capture such intense storms in the Zijingguan catchment accurately. For storm events 1 and 2, the WRF and the coupled WRF/WRF-Hydro systems both exhibited nearly identical spatial patterns in cumulative rainfall distributions. The simulations for these events aligned more closely with cumulative rain gauge observations compared to Events 3 and 4. The WRF and coupled WRF/WRF-Hydro models effectively captured the storm centers of Events 1 and 2. However, for storm Event 3, some areas with high rainfall accumulations within the catchment were not accurately

represented by the WRF and coupled WRF/WRF-Hydro models compared to observations.

| Event 1 | Event 1(a) | Event 1(b) | Event 1(c) |
| Event 2 | Event 2(a) | Event 2(b) | Event 2(c) |
| Event 3 | Event 3 (a) | Event 3 (b) | Event 3 (c) |




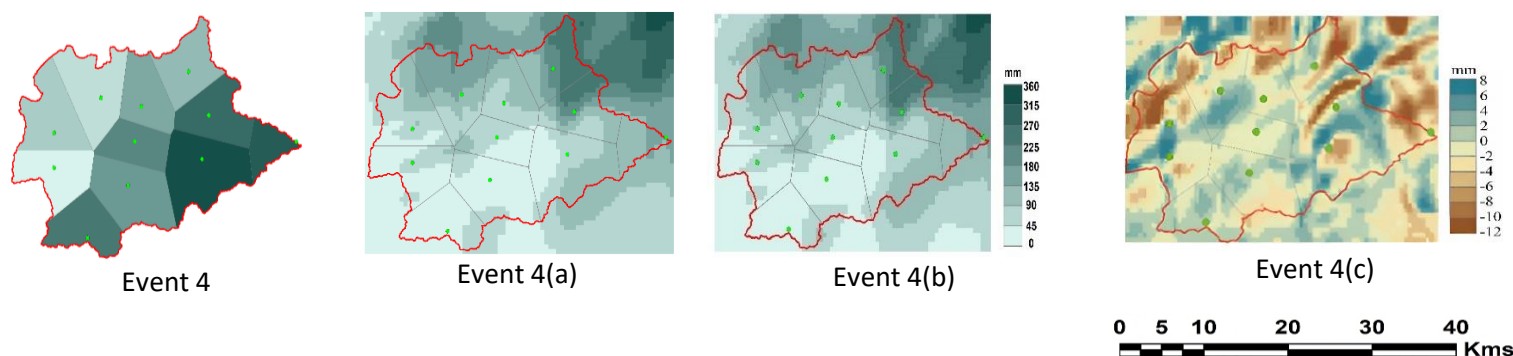

**Figure 9. The 24h accumulative Spatial distributions of the four storm events in the study catchments: gauge observation, (a) WRF model, (b) coupled WRF/WRF-Hydro, (c) Coupled WRF/WRF-Hydro minus WRF.**

## 5. Discussion

The WRF atmospheric model, when coupled with hydrological models, significantly enhances the representation of
precipitation and its impact on hydrological processes. The model's capability to account for localized variations improves
accuracy in capturing spatial heterogeneity, thereby contributing to its effectiveness in various scenarios (Liu et al., 2021).
The results of the coupled WRF/HEC-HMS and WRF/WRF-Hydro systems highlight intriguing findings in flood simulation
within mountainous catchments of Northern China. These coupled models exhibit distinct characteristics in effectively
capturing the complex interplay between precipitation and runoff. The coupled WRF/WRF-Hydro, a fully distributed system,
perform better for extreme storm events, as in the case of storm events 2 and 4. However, this coupled system performed
poorly for storm events that exhibited faster surface runoff recession, exemplified in storm events 1 and 3. This may be
because of the interaction between the land surface and rain-runoff generation, occurring within the WRF-Hydro at a shorter
integration time step. This increases the volume of infiltrated precipitation, leading to higher soil moisture and a decrease in
runoff production (Cuntz et al., 2016). On the other hand, the coupled WRF/HEC-HMS model, a simpler semi-distributed
model, performs better for prolonged floods, as in the case of storm events 1 and 3, as it can adapt to the rapid recession
process. Nonetheless, the coupled WRF/HEC-HMS model may have oversimplified the hydrological processes in some
storm events, resulting in less accurate predictions, particularly in extreme conditions such as events 2 and 4.

The performance error of the lumped system reflects the uncertainty inherent in the hydrological model. The lumped HEC-
HMS is driven by observed data, representing a system with the "true" inputs of the studied catchments. Adopting the
lumped HEC-HMS model allows us to assess the influence of rainfall errors on the coupled systems. The comparison
between the lumped HEC-HMS model driven by observed rainfall and the coupled WRF/HEC-HMS model reveals that the
error is primarily attributed to simulated WRF rainfall. The model structure also has some influence (lumped > gridded),
although this influence is negligible compared to the simulated rainfall error. The results show minimal differences in




average model performances between the lumped HEC-HMS and the coupled WRF/HEC-HMS model. The simulation results of the coupled WRF/HEC-HMS and WRF/WRF-Hydro modeling systems are influenced by the uncertainties in the hydrological models and the driven data. Previous studies have shown that the error in the WRF output rainfall, which depends on the quality of the driven data (i.e., FNL data), necessarily causes parallel uncertainty in hydrological forecasts(Merino et al., 2022). We analyzed the temporal and spatial errors inherent in the simulated WRF output rainfall,
influencing the simulation results of the coupled systems. The temporal variations scrutinize how well the WRF model captures the timing and intensity of rainfall events, providing a comprehensive understanding of its predictive capabilities in the study catchments. The most significant temporal variation between the 24-hour accumulations of observed and simulated rainfall was found in storm Event 4, characterized by the highest rainfall intensity. Additionally, we assess the 24-hour accumulations of rainfall spatial variation, identifying regions where the models may struggle to predict precipitation
patterns accurately in the study catchments. The most significant spatial variation was found in Event 3, where both the WRF and coupled WRF/WRF-Hydro systems failed to capture areas with high rainfall accumulations within the catchment accurately compared to the observations.

Even though there are some uncertainties in the coupled atmospheric-hydrological systems, it is evident that a larger
magnitude of error is attributed to the WRF output rainfall. This indicates that the primary factor influencing the overall accuracy of the coupled systems is the accuracy of simulated WRF rainfall. To enhance simulation rainfall in small and medium-scale mountainous catchments, such as those in Northern China, here are some recommendations: the observed rainfall can be used to correct the simulated rainfall, implementing radar data assimilation in Numerical Weather Prediction (NWP), integrating the simulated rainfall with radar quantitative precipitation forecasts (QPF) or quantitative precipitation
estimates (QPE), etc. It should be noted that the unique topographic and climatic features of our study area compound the inherent uncertainty in simulations. Coupling the WRF atmospheric model with hydrological models introduces additional challenges, such as model parameterization, spatial static data, downscaling resolution, and integration time step, which can collectively influence simulation outcomes. However, the general conclusions of this study aim to provide valuable insights into the performance and potential enhancements of this modeling approach in the face of complex topographical and
meteorological conditions.

## 6. Conclusions

This study coupled the simi-distributed  HEC-HMS and the fully distributed WRF-Hydro models with a 1 X 1 km rainfall output from the WRF model and the lumped HEC-HMS using the observed gauge precipitation. A comparative analysis of the forecast process from the coupled hydrological systems is carried out for storm events with different characteristics. The
lumped HEC-HMS model is adopted as a benchmark to compare and analyze the level of uncertainty in the coupled



WRF/HEC-HMS model system. Also, the error in the simulated WRF output rainfall was analyzed. From the results, we concluded that:

- The coupled WRF/HEC-HMS system performed better for prolonged ( long duration ) storm events and obtained an optimal performance for uniformly distributed storm events in both spatial and temporal (event 1), as it adapts to the rapid recession process, but it did not adequately capture the magnitude of the storm events as it has a larger flow peak error.

- The coupled WRF/WRF-Hydro performed better for shorter-duration floods with higher flow peaks, as it can adapt to the forecasting of flash floods but performs poorly as uniformity decreases. This might be because of model complexity.

- The performance of the lumped model driven by the gauge observed rainfall indicates some uncertainty in the hydrological models when compared with the coupled WRF/HEC-HMS, but a larger magnitude error was found in the WRF output simulated rainfall.

The conclusions from this study verify the notion that the coupled atmospheric-hydrological modeling systems are influenced by the rainfall simulation accuracy and complexity of the hydrological model. We hope this study will encourage further research on improving the simulated rainfall and hydrological models to verify the conclusions of this study.

**Data availability**

The hydrological data used in this study are provided by the State Key Laboratory of Simulation and Regulation of Water Cycle in River Basin, China Institute of Water Resources and Hydropower Research, Beijing. The Final Operational Global Analysis meteorological data (FNL) is available at http://rda.ucar.edu/datasets/ds083.2/). (last access: 25 January 2023) (NCAR, 2023). Access to the 30 m digital elevation model (DEM) can be requested through the website: **http://www.gscloud.cn/sources/?cdataid=302&pdataid=10**. (last access: 25 January 2023) (Geospatial Data cloud, 2023).

**Author contribution**

All authors contributed to the study's conception and design. Research ideas and conceptualization were proposed by JL, YW, and SUJJ. Material preparation, data collection, and analysis were performed by SUJJ and YL. The the first draft of the manuscript was written by SUJJ and JL. And SUJJ and JL did figure production, calculation and editing. All authors read and approved the final manuscript.

**Competing interests**

The authors declare that they have no competing interests..



**Financial support**

This research was supported by the National Natural Science Foundation of China-Regional Innovation and Development Joint Fund (U23A2001) and the National Natural Science Foundation of China (51822906).

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
