# Peer review of "COUPLING WRF WITH HEC-HMS AND WRF-HYDRO FOR FLOOD FORECASTING IN TYPICAL MOUNTAINOUS CATCHMENTS OF NORTHERN CHINA"

_Natural Hazards and Earth System Sciences, 2024_

## Referee Comment (RC2)

**General comments**

The paper evaluates the performance of coupled atmospheric-hydrological modeling systems in order to improve the simulation of flood events. Precisely, it compares 1) the performance of coupling two hydrological systems (fully distributed WRF-Hydro and a simi-distributed Hydrological Engineering Center-Hydrological Modeling System (HECHMS)) with WRF model for four rainfall events in two mountainous areas in China, and 2) the lumped HEC-HMS using the observed gauge precipitation with the simi-distributed HEC-HMS using a 1 x 1 km rainfall output from the WRF model. They concluded that the accuracy of the simulation of the rainfall and the model's complexity influenced the performance of the coupling atmospheric and hydrological models. The manuscript offers an additional contribution to understanding the advantages and disadvantages in the selection of the hydrological model. However, the paper must be more organized in some sections. Thus, I recommend a minor revision of the manuscript before it can be accepted for publication in NHESS.

**Specific comments**

1.  The main objectives and the motivation of this paper must be more clearly explained in the introduction. A reconstruction of the introduction is needed.
2.  The authors must add a paragraph at the end of the introduction that explains the structure of this paper.
3.  The authors must write information in Section 2 about the origin of the gauged data, the total number of stations at the two catchments, the hydrological data, and the initial boundary conditions for WRF.
4.  A small paragraph about the atmospheric circulation and the structure / scale of the selected events will be helpful.
5.  The authors must add a table with the WRF physics schemes used and the Land surface model. Are there previous sensitivity studies that used this options in similar simulated events? Please justify the selection.
6.  More details about FNL data are needed.
7.  The authors must elaborate on the calibration process of WRF-Hydro. Which method did you use, and why? What data did they use for the calibration?
8.  A discussion that would put their results in a comparative context with other studies is missing.
9.  Discuss about the potential of exploiting the presented results under operational forecasting applications, as those presented, for example, by Giannaros et al. (2021) and Varlas et al. (2024).

    https://www.mdpi.com/2571-9394/3/2/26

    https://www.mdpi.com/2073-4433/15/1/120

10. A paragraph about the limitations of this study and if there are uncertainties that impact its results must be added.

**Technical corrections**

1.  Line 64. Which NWP model?
2.  Line 64-65. Please rephrase

3.  Line 67. Please elaborate on the results.

4.  Line 70. Which are the studies that consider the difference?

5.  Line 75. Use capital letters on "valiya veettil et al".

6.  Line 85. Please be more clear about the role of examining the lumped HEC85 HMS model with observations.Line 89. Please give reference to Figure 1.

7.  Line 95. Please elaborate on the results of coupling WRF and WRF-Hydro.

8.  Line 122. Please correct "2219km2" and "1760km2".

9.  Line 128. Which is the strom season?

10. Line 120. It must be "2.1"?

11. Line 141. Is there a reference for the coefficient of variance?

12. Line 164. Reference?

13. Line 167. A title on the x-axis is missing in Figure 2.

14. Line 173. Which version of WRF is used?

15. Line 187: A figure with the WPS domain configuration will be helpful.

16. Line 197.  The information about WRF-Hydro must be reduced.

17. Line 242. The authors must explain how they compute the catchments' routing grids.

18. Line 254. There is no need for figure 3.

19. Line 255. The information about HEC-HMS model must be reduced.

20. Line 303. References?

21. Line 325. Reference?

22. Line 481. This section looks like a summary. The authors must highlight the strong points of the study.

---

## Author Comment (AC1)

**Responses by authors**

We thank Referee #1 for the valuable comments and suggestions which will improve the quality of the manuscript. Detailed responses are provided to your questions. The blue text shows our response, updates that will be incorporated in the manuscript are highlighted in red, and the black text shows the referee's comments.

The manuscript follows a logical order and is generally well-written. I put forward some minor revision suggestions and comments for the authors to consider. I hope that the authors can deal with the comments seriously and make detailed revisions through more in-depth analysis.

Specific comments:

Point 1: Line 446-447. The manuscript mentions that the coupled WRF/HEC-HMS model simplifies hydrological processes for storm events 2 and 4. Could the authors elaborate on the underlying reasons for this limitation?

Response 1: The WRF/HEC-HMS model simplifies several hydrological processes compared to the WRF/WRF-Hydro system. Specifically, the treatment of surface runoff generation, infiltration, evapotranspiration, and routing processes within the HEC-HMS framework involves several assumptions and parameterizations that differ from the more detailed WRF/WRF-Hydro model. For instance, the HEC-HMS model often uses simplified algorithms for surface runoff, assuming uniformity in parameters such as soil properties and land use, which may not capture the spatial variability as effectively as the distributed approach in WRF/WRF-Hydro. These simplifications can significantly impact model accuracy, especially during extreme conditions like storm events 2 and 4. The lumped parameter approach in HEC-HMS might not represent rapid hydrological changes across different sub-basins, leading to less accurate predictions of peak flows and runoff volumes.

To address the reviewer's request for specific details, we updated the following information in the revised manuscript:

"The lumped parameter approach in HEC-HMS, i.e., assuming uniformity in parameters such as soil properties and land use might not be able to present the rapid hydrological changes across different sub-basins."

Point 2: Line 471-475. The manuscript suggests several recommendations for enhancing rainfall simulation, such as using observed rainfall for correction and integrating radar data assimilation.

I suggest the authors add citations of research where these recommendations have been validated or tested.

Response 2: Citations of some of the recommended suggestions for improving WRF rainfall simulation have been added to the manuscript:

"(Vendrasco et al., 2016; Tong et al., 2016; Liu et al., 2021)".

Liu, Y., Liu, J., Li, C., Yu, F., and Wang, W.: Effect of the Assimilation Frequency of Radar Reflectivity on Rain Storm Prediction by Using WRF-3DVAR, https://doi.org/10.3390/rs13112103, 2021.

Tong, W., Li, G., Sun, J., Tang, X., and Zhang, Y.: Design Strategies of an Hourly Update 3DVAR Data Assimilation System for Improved Convective Forecasting, Weather Forecast., 31, 1673–1695, https://doi.org/https://doi.org/10.1175/WAF-D-16-0041.1, 2016.

Vendrasco, E. P., Sun, J., Herdies, D. L., and Frederico de Angelis, C.: Constraining a 3DVAR Radar Data Assimilation System with Large-Scale Analysis to Improve Short-Range Precipitation Forecasts, J. Appl. Meteorol. Climatol., 55, 673–690, https://doi.org/https://doi.org/10.1175/JAMC-D-15-0010.1, 2016.

Point 3: Line 17 and 83 "simi-distributed" should be corrected to "semi-distributed".

Response 3: "Simi-distributed' has been corrected to "semi-distributed".

Point 4: Line 120. "Daqing River basin" should be capitalized as "Daqing River Basin" since it is a proper noun.

Response 4: "Daqing River basin" has been capitalized to "Daqing River Basin".

Point 5: Line 122. "Fuping (2219km2)" and "Zijingguan (1760km2)" should have spaces between the numbers and the unit "km²".

Response 5: The space between the numbers and unit has been added "Fuping (2219 km2)" and "Zijingguan (1760 km2)".

Point 6: Line 125.  "Zijigguan" should be corrected to "Zijingguan".

Response 6: "Zijigguan" has been corrected to "Zijingguan".

Point 7: Line 138. "events happened" can be changed to "events occurred".

Response 7: "events happened" has been changed to "events occurred".

Point 8: Line 168. "Figure 2, subfigure for Event 2" is non-uniform compared to the others and should be replaced.

Response 8: "Figure 2, subfigure for Event 2" has been replaced and all subfigures are now uniform.

[Figure]

Figure 2. The rainfall-runoff observations of the four 24-hour storm events.

Point 9: Line 185. "WRf" should be corrected to "WRF".

Response 9: "WRf" has been corrected to "WRF".

Point 10: Line 189. "sub-watershed" should be corrected to "sub-catchment" for consistency with previous terminology in the manuscript.

Response 10: "sub-watershed" has been corrected to "sub-catchment" for consistency.

Point 11: Line 191. "50hPa top-layer pressure" should have spaces between the number and the unit "hPa" for clarity.

Response 11: the spaces between the number and the unit have been added "50 hPa".

Point 12: Line 256. "Hydrologic" should be "Hydrological" for consistency.

Response 12: "Hydrologic" has been changed to "Hydrological" for consistency.

Point 13: Line 266. "Metrologic data" should be corrected to "Meteorologic data"

Response 13: "Metrologic data" has been corrected to "Meteorologic data".

Point 14: Lines 371 and 394 "storm Event" should be capitalized as "Storm Event".

Response 14: "storm Event" has been capitalized as "Storm Event".

Point 15: Line 404. "as shown in Table 2, has a better simulation result". This phrase seems to be incorrectly placed, disrupting the flow of the sentence, and should be corrected as "(as shown in Table 2), has a better simulation result".

Response 15: The phrase "as shown in Table 2, has a better simulation result" has been corrected as "(as shown in Table 2), has a better simulation result".

Point 16: Line 440. "perform" should be "performs".

Response 16: "perform" has been corrected to "performs".

Point 17: Line 482. "simi-distributed" should be "semi-distributed".

Response 17: "simi-distributed" has been changed to "semi-distributed".

Point 18: Line 484. "is carried out" should be "are carried out".

Response 18: "is carried out" has been corrected to "are carried out".

Point 19: Line 490. "process" should be "processes".

Response 19: "process" has been changed to "processes".

---

## Author Comment (AC2)

**Responses by authors**

We thank Referee #2 for the valuable comments and suggestions which will improve the quality of the manuscript. Detailed responses are provided to your questions. The blue text shows our response, updates that will be incorporated in the manuscript are highlighted in red, and the black text shows the referee's comments.

The manuscript offers an additional contribution to understanding the advantages and disadvantages in the selection of the hydrological model. However, the paper must be more organized in some sections. Thus, I recommend a minor revision of the manuscript before it can be accepted for publication in NHESS.

Specific comments

1. The main objectives and the motivation of this paper must be more clearly explained in the introduction. A reconstruction of the introduction is needed.

Response 1: We have revised the introduction to more clearly articulate the main objectives and motivation of our study. The revised introduction now emphasizes the significance of flood forecasting in mitigating the impacts of floods, the advancements in coupling hydrological models with high-resolution NWP models, and the importance of considering different complexities in hydrological modeling systems. We have also explicitly stated the main objective of our study, which is to evaluate the potential of coupling the WRF model with different hydrological modeling systems to improve flood forecasting accuracy. We believe these changes address your concerns and provide a clearer context and motivation for our research.

Here are the sentences and paragraphs that address the reviewer's comments clarifying the main objectives and motivation, Line 37-40, Line 80-81, and Line 83-85 respectively:

"Flood forecasting is essential to mitigate the impact of floods by providing timely warnings and enabling proactive measures that help safeguard lives, property, and infrastructure in vulnerable areas. Improving the ability to predict flood risks ahead of time is crucial for promoting forecast accuracy."

"Extreme weather events, particularly intense storms, pose significant challenges for hydrological modeling due to their complex interactions with surface and subsurface processes."

"The main objective of this study is to evaluate the potential of coupling the mesoscale numerical weather prediction model, i.e., the weather research and forecasting (WRF) model, with different hydrological modeling systems to improve the accuracy of flood forecasting."

2. The authors must add a paragraph at the end of the introduction that explains the structure of this paper.

Response 2: We have added a paragraph at the end of the introduction that outlines the structure of the paper. This paragraph provides a brief overview of the key analyses performed in the study, including the evaluation and comparison of the coupled systems' performance, the assessment of model uncertainty, and the analysis of errors and uncertainties in the atmospheric-hydrological coupling systems. This addition aims to guide the reader through the organization of the paper and highlight the main aspects of our analysis.

The added paragraph at the end of the introduction explains the structure of this paper, Line 120-126:

"The analysis in section 4 is structured as follows:

- Section 4.1: Evaluate and compare the performance of the coupled systems, i.e., WRF/WRF-Hydro and WRF/HEC-HMS, with different complexities.

- Section 4.2: Evaluate and compare the performance of the coupled WRF/HEC-HMS and lumped HEC-HMS model driven by observed rainfall to analyze the model uncertainty.

- Section 4.3: Evaluate and analyze the error of the WRF model output rainfall and its resulting uncertainty in the atmospheric-hydrological coupling systems."

3. The authors must write information in Section 2 about the origin of the gauged data, the total number of stations at the two catchments, the hydrological data, and the initial boundary conditions for WRF.

Response 3: Origin of Gauged Data: The gauged data used in this study were obtained from the Ministry of Water Resources of the People's Republic of China, which maintains a network of hydrological monitoring stations across the region. These data include measurements of precipitation and streamflow, collected at regular intervals.

The following sentence has been added to section 2.2, Line 152-153 to provide information about the origin of the data:

"The gauged rainfall and flow data was provided by the Ministry of Water Resources of the People's Republic of China."

Number of Stations: The following paragraph has been added to section 2, Line 134-136 to provide information about the total number of stations at the two catchments:

"The Fuping catchment has a total number of 8 gauged stations and the Zijinguan catchment has a total number of 11 gauged stations as shown in Figure 1. Hydrological stations measured the flow at the outlets of the two catchments."

This network provides comprehensive spatial coverage, allowing for detailed analysis of hydrological processes in the area.

4. A small paragraph about the atmospheric circulation and the structure/scale of the selected events will be helpful.

Response 4: A small paragraph about the atmospheric circulation and the structure/scale of the selected events have been added in section 2.2 Line 150-154:

"Storm events in the Fuping and Zijingguan catchments in the Daqinghe Basin are driven by the East Asian Monsoon, bringing moist air and intense rainfall. These storms often form into mesoscale convective systems (MCS), large clusters of thunderstorms with sustained heavy rain, leading to rapid river rises and potential flooding."

5. The authors must add a table with the WRF physics schemes used and the Land surface model. Are there previous sensitivity studies that used these options in similar simulated events? Please justify the selection.

Response 5: The following paragraph and table with the WRF physics schemes and Land surface model used have been added in section 3.1 Line 193-199. And yes, there are previous sensitivity studies that have utilized these options. The selection of these options is justified based on prior studies conducted in the same study area and under similar conditions, which are cited in our manuscript:

The performance of the WRF model largely depends on the parameterization schemes, which can be effective for some storm events but not for others (Liu et al., 2013). Due to the difficulty in determining the best schemes for future storms, these are often pre-set in operational uses (Liu et al., 2015). In this study, we utilize the most widely used physical parameterizations for northern China. Details of the parameterizations that significantly influence precipitation generation are provided in Table 4 and further elaborated by (Tian et al., 2017).

**Table 3. Main WRF Model physical schemes used in this study**

| Parameterization | | Chosen option | | Reference |
|---|---|---|---|---|
| Microphysics scheme | | Lin | | (Lin et al., 1983) |
| Longwave radiation | | Rapid Radiative Transfer Model (RRTM) | | (Mlawer et al., 1997) |
| Shortwave radiation | | Dudhia | | (Dudhia, 1989) |
| Land surface scheme | | Noah | | (Chen and Dudhia, 2001) |
| Planetary boundary layer | | Yonsei University (YSU) | | (Hong et al., 2006) |
| Cumulus convection | | Kain-Fritsch (KF) | | (Kain, 2004) |

6. More details about FNL data are needed.

Response 6: More details about the FNL data have been added in section 3.1 Line 204-208:

The initial boundary conditions for simulation are derived from the 1˚x1˚ FNL driving data at 6-hour intervals, with the integration time step set at 6 seconds (Zhu et al., 2022). The FNL data is the Final Operational Global Analysis meteorological data, which is provided by the National Centers for Environmental Prediction (NCEP) (available: http://rda.ucar.edu/datasets/ds083.2/).

7. The authors must elaborate on the calibration process of WRF-Hydro. Which method did you use, and why? What data did they use for the calibration?

Response 7: The calibration process of the WRF-Hydro model has been elaborated stating the calibration method including the parameters considered and reasons. Citing previous parameter sensitivity studies done in our study area. The following paragraph has been added in section 3.4, Line 306-310 which addresses the reviewer's comments:

"When calibrating the WRF-Hydro model, we employed manual calibration by carefully considering several crucial parameters that have the potential to impact flood forecasting accuracy significantly. These parameters were identified through prior research on parameter sensitivity analysis conducted in the study region (Liu et al., 2021). Because of the limited data in our study area, a systematic adjustment of model parameters to minimize discrepancies between observed and simulated streamflow was done for each storm event to enhance model accuracy."

8. A discussion that would put their results in a comparative context with other studies is missing.

Response 8: We have now added a discussion that puts our results in a comparative context with other studies in the discussion section, Line 477-483, which compares our results with those of previous studies that have utilized similar modeling approaches and study conditions. This comparative analysis highlights the advancements and unique contributions of our work in improving flood forecasting accuracy through the coupling of the WRF model with different hydrological modeling systems:

"In comparison with previous studies, our results show significant improvements in flood forecasting accuracy by coupling the WRF model with different hydrological systems. This aligns with (Jasper et al., 2002) on NWP model development, (Bartholmes and Todini, 2005) on high-resolution NWP data benefits, and (Cattoën et al., 2016) on forecast reliability. Our findings also support (Li et al., 2017) on decreased accuracy with longer lead times, mitigated here by high-resolution models. By integrating fully distributed and semi-distributed models, we extend the work of (Ming et al., 2020) and (Chen et al., 2020) who demonstrated the advantages of coupled NWP and hydrodynamic models."

9. Discuss about the potential of exploiting the presented results under operational forecasting applications, as those presented, for example, by Giannaros et al. (2021) and Varlas et al. (2024).

https://www.mdpi.com/2571-9394/3/2/26

https://www.mdpi.com/2073-4433/15/1/120

Response 9: We have added a brief discussion in Line 483-485 on how our findings can be applied in operational forecasting, inspired by the work of Giannaros et al. (2021) and Varlas et al. (2024).

"Similar to the research of (Giannaros et al., 2021) and (Varlas et al., 2024)), our approach shows that high-resolution NWP data integrated with hydrological models can enhance real-time flood prediction. Implementing these coupled systems in operational frameworks can provide more accurate and timely flood warnings."

10. A paragraph about the limitations of this study and if there are uncertainties that impact its results must be added.

Response 10: We have revised the discussion section and added a paragraph that addresses the limitations, which focus on model configurations, parameterizations, meteorological input

uncertainties, and coupling approaches. This provides a clearer account of the challenges and uncertainties in the modeling process. Section 5 Line 483 – 493:

"It should be noted that the unique topographic and climatic features of our study area compound the inherent uncertainty in simulations. Coupling the WRF model with hydrological models introduces additional challenges, such as model parameterization, spatial static data, downscaling resolution, and integration time step, all of which can significantly influence simulation outcomes. Furthermore, uncertainties in meteorological inputs, such as precipitation forecasts, and the choice of model coupling approaches, such as the one-way coupling used in this study, may affect the reliability of flood predictions. However, the general conclusions of this study aim to provide valuable insights into the performance and potential enhancements of this modeling approach in the face of complex topographical and meteorological conditions."

Technical corrections

1. Line 64. Which NWP model?

Response 1: The "WRF (Weather Research and Forecasting)" model was used as the NWP (Numerical Weather Prediction) model and it has been corrected.

2. Line 64-65. Please rephrase

Response 2: The sentence has been rephrased:

"The system provided a 34-hour lead time based on weather forecasts available 36 hours in advance."

3. Line 67. Please elaborate on the results.

Response 3: The following revised sentence provides a clearer and more concise elaboration on the results of (Patel and Yadav, 2023):

"Their results demonstrated the effectiveness of these coupled systems in accurately predicting reservoir inflows in the Sabarmati River basin in India."

4. Line 70. Which are the studies that consider the difference?

Response 4: The sentence has been clarified and improved for better understanding:

"Although recent studies have been conducted to improve flood forecasting by coupling NWP models with hydrological models, few have addressed the implications of choosing between

fully distributed and semi-distributed models of varying complexities in constructing these coupling systems."

5. Line 75. Use capital letters on "valiya veettil et al".

Response 5: "Valiya Veettil et al" has been capitalized.

6. Line 85. Please be more clear about the role of examining the lumped HEC-HMS model with observations. Line 89. Please give reference to Figure 1.

Response 6: The following revised paragraph now clearly explains the role of the lumped HEC-HMS model in the study and references Figure 1, Line 87-90:

"Additionally, the lumped HEC-HMS model was adopted using observed gauge precipitation as a benchmark to test the rainfall input uncertainty. This approach examines the effectiveness of the HEC-HMS model in replicating observed conditions at the gauge locations as shown in Figure 1."

7. Line 95. Please elaborate on the results of coupling WRF and WRF-Hydro.

Response 7: A paragraph that elaborates on the results and benefits of coupling WRF and WRF-Hydro has been added in Line 100-104:

"This coupling has enhanced flood forecasting accuracy (Senatore et al., 2015) and improved the representation of streamflow dynamics (Ryu et al., 2017). It has demonstrated effectiveness in simulating extreme weather events and their hydrological impacts (Wang et al., 2020; Sun et al., 2020) and in capturing spatial variability in hydrological responses (Quenum et al., 2022; Wang et al., 2022). Additionally, it has proven robust across various climates and geographies, ensuring reliable hydrological predictions (Liu et al., 2023; Naabil et al., 2023)."

8. Line 122. Please correct "2219km2" and "1760km2".

Response 8: The space between the numbers and unit has been added "Fuping (2219 km2)" and "Zijingguan (1760 km2)"

9. Line 128. Which is the storm season?

Response 9: The storm season which is from June to September has been added in Line 137:

"Additionally, during the storm season, typically from June to September, the river undergoes substantial seepage."

10. Line 120. It must be "2.1"?

Response 10: "2." has been changed to "2.1"

11. Line 141. Is there a reference for the coefficient of variance?

Response 11: A reference for the coefficient of variance has been added, Line 158: "(Hosking and Wallis, 1997)".

12. Line 164. Reference?

Response 12: A reference has been added, Line 182: "(Du et al., 2016)".

13. Line 167. A title on the x-axis is missing in Figure 2.

Response 13: The X-axis has been added in Figure 2, Line 184:

[Figure]

**Figure 2. The rainfall-runoff observations of the four 24-hour storm events.**

14. Line 173. Which version of WRF is used?

Response 14: The version of WRF has been added: "The WRF model version 3.7 is used in this study."

15. Line 187: A figure with the WPS domain configuration will be helpful.

Response 15: We did not include a WPS domain configuration because the domain size is not large enough to require a detailed visual. Instead, Table 4 effectively conveys all relevant detail parameterizations, in a clear and organized manner. This format minimizes redundancy and ensures clarity, keeping the manuscript focused on essential data while providing a practical reference for understanding the WRF model setup.

16. Line 197. The information about WRF-Hydro must be reduced.

Response 16: The information about the WRF-Hydro model has been reduced.

17. Line 242. The authors must explain how they compute the catchments' routing grids.

Response 17: How both horizontal and vertical catchment's routing grids were computed has been explained, Line 255-258:

"The horizontal routing grids for catchments in the WRF-Hydro model are computed using the Muskingum Cunge method, which handles channel routing with time-varying parameter estimates and neglects the backwater effect (Wang et al., 2022). In this study, the vertical routing process integrates the Noah-MP Land Surface Model (LSM), which includes four soil layers (10 cm, 30 cm, 60 cm, and 100 cm) spanning a 2 m soil column from top to bottom."

18. Line 254. There is no need for figure 3.

Response 18: Figure 3 has been removed.

19. Line 255. The information about the HEC-HMS model must be reduced.

Response 19: The information about the HEC-HMS model has been reduced.

20. Line 303. References?

Response 20: A reference has been added, Line 298: "(Wu et al., 2016)".

21. Line 325. Reference?

Response 21: A reference has been added, Line 324-325: "(Feldman and (U.S.), 2000)".

22. Line 481. This section looks like a summary. The authors must highlight the strong points of the study

Response 22: The conclusion section has been revised highlighting the strong points of our study.

- The coupled WRF/HEC-HMS system exhibits better performance in predicting prolonged storm events with optimal accuracy in uniformly distributed spatial and temporal patterns (e.g., event 1). It effectively adapts to rapid recession processes, despite challenges in accurately capturing flood magnitudes, leading to larger flow peak errors.

- In contrast, the coupled WRF/WRF-Hydro system performs better for shorter-duration floods characterized by higher flow peaks, demonstrating strong capability in flash flood forecasting. However, its performance reduces as uniformity in storm events decreases, which might be due to incorrect representation of the spatial rainfall.

- The performance of the lumped model driven by the gauge observed rainfall indicates the uncertainty in the hydrological models when compared with the coupled WRF/HEC-HMS, but a larger magnitude error was found in the WRF output simulated rainfall.